# Pre-Conceptional Anti-Thyroid Antibodies and Thyroid Function in Association with Natural Conception Rates

**DOI:** 10.3390/ijerph192013177

**Published:** 2022-10-13

**Authors:** Shoko Konishi, Yuki Mizuno

**Affiliations:** Department of Human Ecology, School of International Health, Graduate School of Medicine, The University of Tokyo, Tokyo 113-0033, Japan

**Keywords:** autoimmunity, fertility, iodine, pregnancy, thyroid

## Abstract

Thyroid functioning is an integral part of the endocrine system that supports conception and pregnancy. Frequent consumption of seafood and iodine by Japanese people may adversely affect their thyroid function. Thus, in this study, we aimed to analyze the associations between iodine intake, thyroid hormones, autoimmunity, and natural conception rates in Japanese women trying to conceive their first child. A prospective study of 24 weeks targeted 80 women with no history of infertility, who did not plan to undergo fertility treatment. Concentrations of urinary iodine concentration and thyroid biomarkers in the serum at enrollment were measured. Thirty-five women naturally conceived during the follow-up. The median (inter-quartile range) urinary concentration of iodine was 297 (165, 500) μg/L. Free triiodothyronine (T3) and free thyroxin (T4) negatively correlated with urinary iodine concentrations. Women with anti-thyroid peroxidase (anti-TPO) ≥ 16 IU/mL had lower conception rates (hazard ratio: 0.28, 95% confidence interval, 0.08–0.92) compared with those with anti-TPO < 16 IU/mL, after adjusting for age and coital frequency. None of anti-thyroglobulin antibody, free T3, free T4, or thyroid-stimulating hormone showed significant associations with conception rate after adjusting for age and coital frequency. The negative association between thyroid autoimmunity and natural conception rates warrants further investigation.

## 1. Introduction

Proper functioning of the maternal immune system is essential for a successful pregnancy and childbirth [1]. Autoimmunity, one of the many forms of immune dysfunction, has been linked to infertility and miscarriage in previous studies targeting patients undergoing fertility treatment [2,3,4]. A higher level of pre-conceptional anti-thyroid peroxidase (anti-TPO) antibodies is associated with an increased risk of fetal loss and decreased live birth rates in women undergoing fertility treatment [2]. However, another study found no association between anti-TPO and cumulative pregnancy rates in patients undergoing in vitro fertilization and intracytoplasmic sperm injection [3]. Among patients who conceived with assisted reproductive technology, the clinical miscarriage rate was higher in thyroid antibody- (anti-TPO, anti-thyroglobulin antibody (anti-TG), or both) positive patients than in antibody-negative patients [4]. Thyroid autoimmunity is linked to infertility and miscarriage in euthyroid women [5,6,7,8,9,10], suggesting that thyroid autoimmunity prevails through pathways other than altered thyroid function.

Limited evidence is available regarding the association between thyroid autoimmunity and the natural pregnancy rate. A prospective cohort study targeting women in the United States trying to conceive reported no association between thyroid autoimmunity (TPO or TG) and the probability of pregnancy or miscarriage [11]. The participants had no history of infertility and had one or two prior pregnancy losses [11]. To the best of our knowledge, no other prospective study has examined the association between thyroid autoimmunity and pregnancy probability in women trying to conceive naturally. To examine whether thyroid autoimmunity is associated with a lower pregnancy rate in the general population, it is necessary to conduct a study that targets women without a history of infertility.

High iodine intake is associated with a higher incidence of thyroid autoimmunity [12]. An optimal level of iodine intake is necessary to maintain proper thyroid function [13]. Thus, high and low iodine intake may be associated with a decreased probability of conception through altered thyroid function and autoimmunity. A population-based cross-sectional study targeting pregnant women in China found that women with iodine deficiency (i.e., urinary iodine concentration < 150 µg/L) were more likely to report a time to pregnancy of longer than 13 months [14]. The longer time to pregnancy and lower probability of conception among women with lower iodine intake remained similar when the analysis was limited to women who conceived naturally [14]. In the Chinese study, the median urinary iodine concentrations were 119.6 µg/L in the coastal regions and 147.1 µg/L in the inland regions [14]. Similarly, in a population-based prospective cohort study targeting women in the United States trying to conceive naturally, those with lower urinary iodine concentrations (<50 µg/g creatinine) were less likely to conceive than those with higher iodine levels (≥100 µg/g creatinine) [15]. In the US study, the median (inter-quartile range) urinary iodine concentration was 112.8 (53.6, 216.9) μg/L. However, an important research gap is that none of these studies evaluated thyroid autoimmunity or thyroid function, which may mediate the link between iodine intake and pregnancy rate. Although these previous studies [14,15] suggest a lower pregnancy rate in women with lower iodine intake, it is unknown whether high iodine intake is associated with a lower natural pregnancy rate.

Considering this gap in knowledge, this study evaluated the associations between iodine intake, thyroid autoimmunity and function, and natural pregnancy. We targeted Japanese women trying to conceive their first child, who had never received fertility treatment. Japanese people consume various types of seaweed, and their iodine intake levels are expected to be high [16]. A nationwide survey targeting school children in Japan revealed that the median urinary iodine concentration was equal to or higher than 300 μg/L in 12 of the 46 observed regions, suggesting excess iodine intake [17]. We thus hypothesized that a higher iodine intake might be linked to a higher positivity of thyroid autoimmunity and lower thyroid function, resulting in a decreased probability of natural conception. Two previous studies in Japan in the 1990s reported positive [18] or no associations [19] between urinary iodine concentration and thyroid autoimmunity. This study aimed to (i) examine whether dietary iodine intake is associated with individual variation in thyroid-related biomarkers and (ii) analyze whether pre-conceptional anti-TPO, anti-TG, free triiodothyronine (T3), free thyroxin (T4), and thyroid-stimulating hormone (TSH) levels are associated with natural conception rates in women trying to conceive their first child.

## 2. Materials and Methods

The methods and design of the Baby Machi (meaning “waiting for a baby” in Japanese) study have been described in a previous paper [20]. The inclusion criteria were women aged 20–34 years who were trying to conceive their first child naturally. Couples who had consulted a doctor for fertility treatment or examination were excluded. Participants were enrolled on 15 November 2015, when spot urine and venous blood specimens were collected, and height and weight were measured. Information on coital frequency in the last three months, history of pregnancy, duration since discontinuing contraception, and the last method of contraception was also requested. Follow-up was a maximum of 24 weeks, and was discontinued when pregnancy was confirmed by a medical doctor, if fertility treatment was initiated, or if contraception was restarted.

This study was approved by the Ethics Committee of the Graduate School of Medicine, University of Tokyo [No; 10878-(2)]. The participants provided signed informed consent upon participation.

### 2.1. Laboratory Analyses

Urine and serum specimens were stored at −20 °C and −80 °C, respectively, until laboratory analysis. Creatinine concentrations in the urine specimens were measured using the Jaffe method (Lab Assay Creatinine Kit; FUJIFILM Wako Pure Chemical Corp., Osaka, Japan). Specific gravity (SG) was measured using a pocket refractometer (ATAGO Co., Ltd., Tokyo, Japan). Creatinine and specific gravity measurements were conducted in the laboratory at the Department of Human Ecology, the University of Tokyo.

Urinary iodine (*m*/*z* = 127) concentrations were measured using inductively coupled plasma mass spectrometry (ICP-MS; Agilent 7500ce; Agilent Technologies, Santa Clara, CA, USA). For ICP-MS analysis, a 20-fold dilution with 0.025% tetramethylammonium hydroxide (TMAH; Tama Chemicals Co., Ltd., Kanagawa, Japan) was performed. The diluted samples were filtered through a disposable cellulose acetate membrane filter (Sartorius, Göttingen, Germany). Potassium iodate was used to prepare the working standard solutions at concentrations of 1, 10, and 30 µg/L. Seronorm Trace Elements Urine L-2 (SERO AS, Billingstad, Norway) was used for analytical quality assurance. The author (Y.M.) confirmed that the observed iodine concentrations of the reference material were within the uncertainty range of the reference values with small standard deviations and that the analyses were accurate.

Serum specimens were sent to the laboratory of SRL Co., Ltd., Tokyo, Japan, and the concentrations of anti-TPO, anti-TG, TSH, free T3, and free T4 were determined. All SRL measurements were performed using an electrochemiluminescence immunoassay (ECLIA). The reference values provided by SRL Co., Ltd. were <16 IU/mL for anti-TPO, <28 IU/mL for anti-TG, 0.500–5.00 μIU/mL for TSH, 2.30–4.30 pg/mL for free T3, and 0.90–1.70 ng/dL for free T4.

### 2.2. Statistical Analyses

Descriptive statistics were calculated for the whole sample and those with and without natural conception or pregnancy loss during follow-up. The median (inter-quartile range), proportion, or mean (SD) were calculated for anti-TG, anti-TPO, free T3, free T4, TSH, and urinary iodine concentrations (unadjusted, SG-adjusted, and creatinine-adjusted). Of the participants, 74% and 71% had anti-TG or anti-TPO values equal to or below the lowest limit of <10 IU/mL and <9 IU/mL, respectively (Appendix A). For statistical analyses, the anti-TG and anti-TPO values were dichotomized using the reference values of 28 IU/mL and 16 IU/mL. When anti-TG and anti-TPO concentrations were treated as continuous variables, anti-TG < 10 IU/mL was substituted with 10 IU/mL, and anti-TPO < 9 IU/mL was substituted with 9 IU/mL. Spearman correlation coefficients between urinary iodine concentrations (unadjusted, SG-adjusted, and creatinine-adjusted), anti-TG, anti-TPO, free T3, free T4, and TSH levels were calculated. A Cox regression model was used to assess the association between the probability of natural pregnancy and anti-thyroid antibodies (dichotomized) or thyroid hormones (tertiles). For each explanatory variable, we ran three sets of models. The first set included each thyroid biomarker as an independent variable. The second set of models was adjusted for age as a covariate. The third set of models was further adjusted for coital frequency. BMI was not included in the Cox regression models because it was not associated with the probability of pregnancy [20]. All statistical analyses were performed using R ver. 4.1.1. [21].

## 3. Results

The mean (SD) age of the participants was 29.5 (2.7) years, and the mean (SD) BMI was 20.8 (2.4) kg/m^2^ (Table 1). Condoms were the most common method of contraception. During the 24-week follow-up, 35 women conceived spontaneously. The median (inter-quartile range) of the observed number of menstrual cycles in those who conceived spontaneously was 6.0 (5.0, 7.0) and in those who did not conceive was 2.0 (1.5, 3.5). The median coital frequency was 4.0 (2.5, 8.0) for those who conceived and 2.5 (2.4, 4.0) for those who did not conceive (Table 1). Fifteen (19%) women had ever been pregnant (Table 1), but none of whom had given live birth by the time of enrollment. The proportions of women who had ever become pregnant before enrollment were similar between the two groups, i.e., 18% in the no conception group and 20% in the spontaneous conception group during follow-up (Table 1). The median (inter-quartile range) unadjusted iodine concentration in the urine was 297 (165, 500) μg/L (Table 2). The minimum and maximum unadjusted iodine concentrations were 47 μg/L and 4812 μg/L, respectively. The mean (SD) free T3 and free T4 were 3.1 (0.7) pg/mL and 1.3 (0.2) ng/dL, respectively. Anti-TPO was equal to or higher than 16 IU/mL in 23% of all participants (33% in those who did not conceive and 9% in those who conceived) (Table 2). None of the three women who had a high TPO-Ab level and conceived naturally had early pregnancy loss (Table 2). Free T3 was negatively associated with unadjusted urinary iodine concentrations, whereas free T4 was negatively associated with SG- and creatinine-adjusted urinary iodine concentrations (Table 3). In the bivariate model, a higher anti-TPO level was associated with a lower hazard ratio (HR) for spontaneous pregnancy (HR, 0.24; 95% confidence interval [CI]: 0.07, 0.77) (Table 4). This trend did not change when adjusted for age and coital frequency (Table 4). Anti-TG, free T3, free T4, and TSH levels did not show significant associations with spontaneous pregnancy rates (Table 4).

## 4. Discussion

These results suggest that a higher anti-TPO level may be associated with lower pregnancy rates. The association between anti-TPO and natural conception remained significant, even after adjusting for age and coital frequency. Anti-TG, free T3, free T4, and TSH levels were not associated with the spontaneous pregnancy rate. A median urinary iodine concentration of 297 µg/L suggested that iodine intake levels of this population were above the requirements [22]. Urinary iodine concentrations showed modest negative associations with free T3 and T4 but not with TSH or thyroid autoimmunity.

The lower conception rates found in women with a higher anti-TPO than those with a lower anti-TPO are inconsistent with a previous US study that found no differences in pregnancy rates by thyroid autoimmunity positivity [11]. This US study defined positivity as anti-TG ≥ 115 IU/mL or anti-TPO ≥ 35 IU/mL [11]. In contrast, in our study, higher anti-TG (≥28 IU/mL) and higher anti-TPO (≥16 IU/mL) levels were defined separately based on the clinical reference values provided by the company. The assays used to quantify anti-thyroid antibody levels differ across studies, and the values are not comparable. Anti-TPO concentration as a continuous variable was negatively associated with pregnancy rate, which supports the possibility that there may be some underlying but unknown biological mechanisms linking them. When treated as continuous variables, logged anti-TG was not associated with pregnancy rate (HR, 1.01; 95% CI: 0.75, 1.35), and anti-TPO showed a non-significant negative association with the natural conception rate (HR, 0.42; 95% CI: 0.17, 1.06). A US study [11] included 1228 women with a history of pregnancy loss, and our study showed that only 19% of the women had experienced a pregnancy loss before enrollment. The outcomes examined in the US study included live birth, any pregnancy loss, clinical pregnancy loss, and time to pregnancy tested with the urinary concentration of human chorionic gonadotropin (hCG) [11]. None of these outcomes were associated with the positivity of anti-thyroid antibodies [11]. On the other hand, in the present study, the main outcome was clinical pregnancy, which was negatively associated with anti-TPO. Another important difference between this study and the US study [11] is the relationship between TSH and anti-thyroid antibodies. Although there were no correlations between anti-TG or anti-TPO and TSH in this study, TSH was significantly higher among those with positive versus negative anti-thyroid antibodies in the US study [11].

None of free T3, free T4, nor TSH were associated with conception rate, suggesting that anti-TPO could adversely affect the process directly rather than through altered thyroid hormone levels. It is speculated that anti-TPO can act directly on the oocytes and decrease their quality, lowering their developmental potential [23,24]. However, anti-TPO and anti-TG are both present in follicular fluid [25], and we cannot explain why only anti-TPO, but not anti-TG, was associated with a reduced probability of natural pregnancy in this study population. In addition, the anti-TPO and anti-TG levels were positively correlated (rho = 0.587, *p* < 0.001). Further studies with larger sample sizes are needed to examine the potential differences between anti-TPO and anti-TG levels concerning the probability of natural pregnancy.

The median unadjusted iodine concentration in urine was 297 µg/L in this study, indicating that at a population level, the iodine intake level was above the requirements [22]. The median value is close to the cutoff (300 µg/L) for excessive iodine intake, which is assumed to increase the risk of iodine-induced hyperthyroidism and autoimmune thyroid diseases [22]. However, no significant correlations were found between urinary iodine concentrations and TSH or thyroid antibodies. On the other hand, urinary iodine concentrations were negatively correlated with free T4 and free T3. It is unknown whether the observed correlations reflect the short-term (e.g., several days or weeks) intake of iodine and its impact on thyroid hormone levels or whether it reflects habitual intake of iodine at a longer time scale and its effect on thyroid hormone concentrations. A further study with multiple sampling of urine and blood samples from each participant will be needed to investigate how within-individual fluctuations of iodine intake may affect within-individual variations of T3, T4, and TSH levels across time. The lack of association between urinary iodine concentration and TSH contrasts with a previous study in Japan that found a positive association [18]. This difference may be related to the higher iodine intake in the previous study [18] compared with the present study and the sex of the participants. The present study targeted women; however, most participants in the previous study were men [18]. The proportion of participants with urinary iodine >9525 µg/L (75 µmol/L) ranged between 3.7% and 30.3% in the different regions studied [18]. In the present study, no participant had urinary iodine levels >9525 µg/L. The lack of an association between thyroid autoimmunity and urinary iodine is consistent with a cross-sectional study in Japan [18]. A longitudinal study in China found that the incidence of thyroid autoimmunity over a five-year follow-up was higher in areas with adequate or excessive iodine intake than in areas with mildly deficient iodine intake [12].

None of the three women with elevated TPO-Ab levels who had conceived naturally experienced early pregnancy loss. The sample size was too small to conclude that there was no increase in pregnancy loss due to TPO-Ab. Participants were instructed to test for pregnancy using the hCG-based kit twice per cycle [20]; as such, we could detect early pregnancies. Thus, it is unlikely that early pregnancy losses caused a lower pregnancy rate. Instead, the present results suggest that a higher anti-TPO level is associated with a lower pregnancy rate. Mechanisms linking anti-TPO and lower conception rates are unknown, but potential mechanisms include lower-quality oocytes [23], linked to lower probabilities of fertilization, implantation, and embryogenesis. Both anti-TPO and anti-TG have been detected in follicular fluid [25].

The small sample size was a limitation of this study. Anti-TPO was associated with a lower pregnancy rate; however, the 95% CIs of the HRs were wide. The small sample size did not allow us to separately analyze the combined and independent effects of higher anti-TPO and anti-TG levels. Another limitation is that we did not obtain information on the birth outcome of the pregnancy observed during the 24-week follow-up. Therefore, we could only calculate the probability of natural pregnancy but not of live birth. It is possible that the present participants may have lower pregnancy rates compared with the general population in Japan. While 44% of the present participants reported clinical pregnancy during the 24-week follow-up, the cumulative pregnancy rate after six months of discontinuing contraception was estimated to be 57%, 54%, and 41% for women aged 27–29, 30–32, and 33–35 years, respectively [26]. A follow-up period of 24 weeks is sufficient to examine the association of thyroid antibodies and function with the natural pregnancy rate, especially considering that within two years of enrollment, 56% of women had sought fertility treatment [20]. The use of urinary iodine concentration as a measure of iodine intake for each individual is another limitation of this study. The median urinary iodine concentration is used to assess the iodine intake level of a population [22] but not of an individual. Future studies with a larger sample size evaluating outcomes of pregnancy rate, pregnancy loss, and livebirth rate are warranted to confirm this study’s findings. The strength of this study was its prospective design, with frequent pregnancy testing using hCG test kits. The participants were limited to women trying to conceive their first child spontaneously, and none had given birth or had a history of infertility. Therefore, the present results suggest that thyroid autoimmunity may, to some extent, explain the variability in fecundability across couples [27] trying to conceive their first child.

## 5. Conclusions

Eighty women trying to conceive their first child were followed up for a maximum of 24 weeks. Urinary iodine concentrations were negatively associated with free T3 and T4 but not with TSH or thyroid antibodies. A higher level of pre-conceptional anti-TPO was associated with a decreased probability of natural conception. Free T3, free T4, TSH, and anti-TG were not associated with conception, suggesting that anti-TPO may be linked to a decreased natural conception rate via pathways other than decreased thyroid hormones. Future studies with larger sample sizes are warranted to confirm the present findings.

## Figures and Tables

**Table 1 ijerph-19-13177-t001:** Basic characteristics of all participants as well as those who did and did not achieve spontaneous conception during follow-up. Mean (SD), median (inter-quartile range) or proportion (*n*).

		Spontaneous Conception	
	All (*n* = 80)	No (*n* = 45)	Yes (*n* = 35)
Age (y)	29.5 (2.7)	29.9 (2.8)	28.9 (2.6)
Height (cm)	159.8 (5.5)	160.6 (4.8)	158.9 (6.2)
Weight (kg)	53.1 (7.3)	54.3 (6.3)	51.6 (8.2)
BMI (kg/m^2^)	20.8 (2.4)	21.0 (2.0)	20.4 (2.8)
Ever pregnant before enrollment	19% (15)	18% (8)	20% (7)
Ever smoked	16% (13)	13% (6)	20% (7)
University degree or higher	78% (62)	76% (34)	80% (28)
Coital frequency ^a^	3.3 (2.5, 8.0)	2.5 (2.5, 4.0)	4.0 (2.5, 8.0)
Duration without contraception at enrollment (month)	3 (6, 18)	3 (6, 24)	3 (6, 13)
Last method of contraception ^b^			
Condom	74% (59)	73% (33)	74% (26)
Pill	16% (13)	16% (7)	17% (6)
Interruptus	24% (19)	29% (13)	17% (6)
Calendar	10% (8)	18% (8)	0% (0)
Basal body temperature	3% (2)	4% (2)	0% (0)

^a^ Number of days of sexual intercourse in a month in the past three months reported at enrollment. ^b^ Multiple answers.

**Table 2 ijerph-19-13177-t002:** Thyroid autoimmunity, thyroid function, and urinary iodine concentration for all participants, by spontaneous conception and by early pregnancy loss. Median (inter-quartile range), proportion (*n*), or mean (SD).

		Spontaneous Conception		Early Pregnancy Loss ^a^	
	All (*n* = 80)	No (*n* = 45)	Yes (*n* = 35)	No (*n* = 27)	Yes (*n* = 7)
anti-TG (IU/mL)	10.0 (10.0, 11.3)	10.0 (10.0, 13.0)	10.0 (10.0, 10.0)	10.0 (10.0, 10.0)	10.0 (10.0, 10.0)
anti-TG ≥ 28 IU/mL	16% (13)	13% (6)	20% (7)	22% (6)	14% (1)
anti-TPO (IU/mL)	9.0 (9.0, 10.5)	9.0 (9.0, 19.0)	9.0 (9.0, 9.0)	9.0 (9.0, 9.0)	9.0 (9.0, 9.0)
anti-TPO ≥ 16 IU/mL	23% (18)	33% (15)	9% (3)	11% (3)	0% (0)
free T3 (pg/mL)	3.1 (0.7)	3.2 (0.9)	3.0 (0.3)	3.0 (0.3)	3.0 (0.4)
free T4 (ng/dL)	1.3 (0.2)	1.3 (0.3)	1.3 (0.1)	1.3 (0.1)	1.4 (0.1)
TSH (μIU/mL)	1.9 (1.1)	1.9 (1.1)	1.9 (0.9)	2.0 (1.0)	1.6 (0.7)
Urinary iodine (μg/L)	297 (165, 500)	308 (194, 499)	277 (138, 427)	295 (152, 510)	210 (104, 311)
Urinary iodine (μg/g creatinine)	226 (162, 480)	256 (184, 554)	182 (144, 359)	229 (161, 413)	161 (144, 178)
Urinary iodine (SG-adjusted, μg/L)	259 (186, 541)	284 (211, 569)	242 (171, 442)	244 (181, 481)	181 (166, 241)

^a^ Pregnancy outcome of one woman was unknown. anti-TG: anti-thyroglobulin antibody, anti-TPO: anti-thyroid peroxidase antibody, T3: triiodothyronine, T4: thyroxin, TSH: thyroid-stimulating hormone, SG: specific gravity.

**Table 3 ijerph-19-13177-t003:** Spearman correlation coefficients between urinary iodine concentrations, anti-TG, anti-TPO, free T3, free T4, and TSH.

	(2)	(3)	(4)	(5)	(6)	(7)	(8)
(1) Anti-TG	0.587 ***	0.149	0.124	0.112	−0.118	−0.214	−0.160
(2) Anti-TPO		0.064	0.103	−0.068	0.036	−0.089	−0.060
(3) Free T3			0.385 ***	0.110	−0.258 *	−0.196	−0.179
(4) Free T4				−0.134	−0.196	−0.272 *	−0.322 **
(5) TSH					−0.028	0.116	0.116
(6) Urinary iodine						0.855 ***	0.758 ***
(7) SG-adjusted urinary iodine							0.926 ***
(8) Creatinine-adjusted urinary iodine							

* *p* < 0.05, ** *p* < 0.01, *** *p* < 0.001. anti-TG: anti-thyroglobulin antibody, anti-TPO: anti-thyroid peroxidase antibody, T3: triiodothyronine, T4: thyroxin, TSH: thyroid-stimulating hormone, SG: specific gravity.

**Table 4 ijerph-19-13177-t004:** Hazard ratios (HRs) and their 95% confidence intervals (CIs) of spontaneous pregnancy by thyroid function biomarkers.

Predictor Variables	Bivariate Models	Adjusted for Age	Adjusted for Age and Coital Frequency
anti-TG (IU/mL)			
<28	1.0	1.0	1.0
≥28	1.50 (0.66, 3.44)	1.55 (0.68, 3.56)	1.68 (0.73, 3.88)
anti-TPO (IU/mL)			
<16	1.0	1.0	1.0
≥16	0.24 (0.07, 0.77)	0.24 (0.07, 0.80)	0.28 (0.08, 0.92)
free T3 ^a^ (pg/mL)			
2.14–2.82	1.0	1.0	1.0
2.84–3.14	1.47 (0.66, 3.29)	1.40 (0.62, 3.13)	1.65 (0.72, 3.78)
3.17–7.74	1.14 (0.49, 2.62)	1.14 (0.49, 2.62)	1.19 (0.51, 2.76)
free T4 ^a^ (ng/dL)			
1.05–1.25	1.0	1.0	1.0
1.26–1.37	1.53 (0.64, 3.63)	1.37 (0.57, 3.29)	1.43 (0.06, 3.42)
1.38–3.01	1.98 (0.86, 4.59)	1.89 (0.81, 4.38)	2.03 (0.87, 4.74)
TSH ^a^ (μIU/mL)			
0.005–1.39	1.0	1.0	1.0
1.40–2.17	1.28 (0.54, 3.01)	1.24 (0.52, 2.91)	1.13 (0.48, 2.70)
2.28–5.24	1.76 (0.78, 3.96)	1.72 (0.76, 3.88)	1.52 (0.67, 3.48)

^a^ The first tertiles of each biomarker were set as reference categories. anti-TG: anti-thyroglobulin antibody, anti-TPO: anti-thyroid peroxidase antibody, T3: triiodothyronine, T4: thyroxin, TSH: thyroid-stimulating hormone.

## Data Availability

The data presented in this study are available on request from the corresponding author. The data are not publicly available due to ethical considerations.

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
