# Peer review of "Pre-Conceptional Anti-Thyroid Antibodies and Thyroid Function in Association with Natural Conception Rates"

_ijerph, 2022, doi:10.3390/ijerph192013177_

Round 1
Reviewer 1 Report (Previous Reviewer 1)
The authors have appropriately revised the MS. No further comments.
Reviewer 2 Report (Previous Reviewer 2)
Dear Authors,
Thank you very much for your corrections. I have no further comments.
Reviewer 3 Report (Previous Reviewer 3)
The Authors responded exaustively to the all request , so I belive that the paper can be suitable for publication in the present form
This manuscript is a resubmission of an earlier submission. The following is a list of the peer review reports and author responses from that submission.
Round 1
Reviewer 1 Report
The authors have focussed on the apparent difference in the antiTPO values to explain an apparent difference in conception rates.
1. The study is small and the findings opposite to a much larger study (Plowden et al, cited as reference 11). That publication is slightly misquoted to suggest that only live birth rate was reported, but time to conception was examined as well in that study (table 4) and no difference found between the rate with and without thyroid antibodies.
2. The antiTPO value is dichotomised between greater and less than 9 U/L by ECLIA. The normal range for the SRL assay used is not cited, but from public source data appears to be 34 U/L, so that the chosen value is well within normal range. The biological and analytical validity is therefore questionable. It is unexplained why that value was chosen and the sensitivity of the findings to that value is not examined.
3. The coital frequency is different between the groups and would seem a more likely explanation of the data than the antiTPO values.
4. In the opinion of the reviewer the study needs to be bigger and provide data to rebut the concerns expressed above.
Reviewer 2 Report
Dear Authors,
This is a very interesting study, since the combination of thyroid function parameters, anti-thyroid antibodies and iodine intake at pre-conceptional stage has not been exhaustively studied so far.
The study offers a privileged group of patients, aiming to conceive but without a previous history of infertility. The manuscript is well written, and the discussion contains all the relevant references to put the findings in context.
There only some minor questions that should be addressed:
1. Lines 226-227: The associations found between FT3/FT4 and urinary iodine concentration were both negative, what can be in contradiction with the previous statement (lines 224-226). This finding should deserve some potential explanations at the discussion.
2. Line 247: “lower-quality ovaries”. Do you mean “lower-quality oocytes”??
3. Lines 259-262: I cannot understand this sentence, what kind of intake?
4. Lines 268-271: The paragraph highlights the fact that participants are trying to conceive their first child. However, according to line 208, a 19% of women had already experienced pregnancy losses prior the enrollment. Probably, it would be more accurate to say that they are trying to conceive spontaneously and none of the participants have given birth before. It would also be desirable to clarify if there were any differences between those who had suffered a miscarriage and those who had not.
Reviewer 3 Report
The study addresses a debated and interesting topic concerning the impact of thyroid autoimmunity on fertility.I found the study design appreciable, which explores the impact of antithyroid antibodies and in addition to the population iodine status in a sample of women with no experience of infertility. Certainly the study has some weaknesses in particular related to the size of the sample and the impossibility of assessing the real impact of the presence of antithyroglobulin antibodies on pregnancy rates.On the other hand, by virtue of the general greater pathogenicity of antireoperoxidase antibodies on thyroid tissue, the data, if confirmed, may not be surprising. As the authors themselves indicate there are limits relating to the measurement of ioduria which however are presented for discussion.
Ultimately I judged the study, however reiterating the need for confirmations on much more numerous samples worthy of publication in this form without significant changes